# Contrastive Studies between Laser Repairing and Plasma Arc Repairing on Single-Crystal Ni-Based Superalloy

**DOI:** 10.3390/ma12071172

**Published:** 2019-04-10

**Authors:** Cheng Wang, Qiuliang Li, Xin Zhou, Wenxin Zhu, Runqiu Huang, Zhihao Pan, Kai Chen, Chang He

**Affiliations:** 1Science and Technology on Plasma Dynamics Laboratory, Air Force Engineering University, Xi’an 710038, China; warrant_74@126.com (C.W.); qiuliang_95@163.com (Q.L.); 2State Key Laboratory for Mechanical Behavior of Materials, Xi’an Jiaotong University, Xi’an 710049, China; zwx185@163.com (W.Z.); Richardwong23@163.com (R.H.); zhpangaea@163.com (Z.P.); 3College of Aircraft Engineering, Nanchang Hangkong University, Nanchang 330063, China; lql_kgd@163.com

**Keywords:** Ni-based superalloy, single crystal, laser repairing, plasma arc repairing

## Abstract

Laser repairing and plasma arc repairing experiments on the single-crystal Ni-based superalloy DD407(Ni-7.82Cr-5.34Co-2.25Mo-4.88W-6.02Al-1.94Ti-3.49Ta in wt.%) were carried out in this paper, and the differences in microstructures and mechanical properties varying with depth between the two repairing methods were studied. Comparing the two repairing processes, both the fusion zone can maintain single-crystal epitaxial growth with no significant cracks and have similar fine γ′ precipitates. Nevertheless, the columnar-to-equiaxed transition phenomenon occurred on the top of the fusion zone during the laser re-melting process but was not very obvious on the top surface of the fusion zone during the plasma arc re-melting process. In addition, both the DD407 superalloy conducted with the two repairing processes had a large microhardness and a great Young’s modulus in the fusion zone.

## 1. Introduction

Single-crystal Ni-based superalloys have been widely used in advanced aero-engine turbine blades and vanes, showing excellent stress rupture life, better creep resistance, and superior thermal fatigue resistance because grain boundaries are eliminated, and thus, grain boundary strengthening elements (e.g., low melting point B, C) can be reduced [1,2]. These turbine blades will inevitably be subject to blade tip abrasion and cracks during servicing processes, for reasons of creep elongation and rub-impact under high temperature, high pressure, centrifugal force, and corrosive environments. For blades that only have an abraded part of the blade tip, it will cause great waste and will be economically hard to bear if all the blades are replaced with a new one [3]. Fortunately, the blade tip area can be repaired in some specific circumstances according to the repair manual [4]. In spite of this, the repair process is still difficult for the both the blade shape requirements and epitaxial growth microstructure requirements [5,6,7,8].

Repair attempts started with the use of different welding and cladding processes. Gäumann et al. [9,10] have indicated that a single-crystal deposit can only be produced where conditions that are sufficient for substrate remelting for epitaxial growth and temperature gradient control for columnar dendrite growth are simultaneously achieved. Accordingly, parameters such as relatively low temperature of the substrate, reduced laser beam power, and small laser beam diameter, so as to increase the temperature gradient and decrease the solidification velocity in the melt track, are essential. Ramsperger et al. [11,12] have shown that crack-free CMSX-4(Ni-6.5Cr-9.7Co-0.4Mo-3Re-6.4W-5.6Al-1.0Ti-6.5Ta in wt.%) specimens can be produced by properly controlling the processing temperature field. Rottwinkel et al. [13] have reported that the temperature gradient control of the melt pool is realized by establishing active heat dissipation conditions, and the ratio of preheating temperature to laser power has a significant effect on promoting single-crystal growth and suppressing the stray crystals. Acharya et al. [14] have developed a scanning laser epitaxy (SLE) technique to produce a single-crystal superalloy—CMSX-4—with fully dense, crack-free, and porosity-free deposits. High-resolution scans and tight thermal control are favorable for the fine-grain structure. Basak et al. [15] have further developed the scanning laser epitaxy technique demonstrating the capability of processing uniform deposits of CMSX-4 without any cracks. This method is capable of fabricating dense, crack-free deposits with heights exceeding 1000 μm (single-crystal heights of more than 500 μm), widths exceeding 6000 μm, and lengths exceeding 35,000 μm.

Xu et al. [16] have deposited an Inconel 625 superalloy by using the technique of plasma arc additive manufacturing and investigated the effect of deposition strategy and post-treatment on the microstructure and mechanical properties of as-deposited samples. Lin et al. [17] fabricated Ti-6Al-4V thin wall by continuous plasma arc additive manufacturing and analyzed microstructure’s evolution and mechanical properties. Zhao et al. [18] manufactured GH163(Ni-19Cr-19Co-5.5Mo-0.04C-0.3Al-1.9Ti-0.005B-07Fe in wt.%) samples with the technique of plasma transferred arc-assisted deposition and investigated the effect of solid solution and ageing treatments on γ′ precipitates. The heatsource types and characteristics were considered as very important influencing factors in the columnar grains’ epitaxial growth. However, the contrastive studies between different welding processes on the repair of single-crystal Ni-based superalloys are insufficient. Plasma arc welding or arc welding as well-known processes have the tendency of forming coarse columnar grains, that may be detrimental in engineering alloys welding but may facilitate epitaxial growth in single-crystal repairing. In this paper, the contrasts between laser re-melting and plasma arc re-melting are studied through analyzing the variation of microstructures and mechanical properties versus depth.

## 2. Materials and Methods 

The investigated DD407 superalloy is mainly used as single-crystal turbine blade material on turboshaft engines, which has a nominal composition of Ni-7.82Cr-5.34Co-2.25Mo-4.88W-6.02Al-1.94Ti-3.49Ta in wt.%. A cylindrical single-crystal rod, 13 mm in diameter and 20 cm in length, was directionally solidified under the same process flow of turbine blade production. The growth orientation of a single-crystal rod was along the orientation of [001]. The single-crystal rod was cut using wire electrical discharge machining along the cross-section with 13 mm in diameter and 4 mm in height. Its surface was re-melted using a 2 KW three-axis laser cladding device with a scanning speed of 520 mm/min and defocus distance of +10 mm. The surface of the same single-crystal rod was re-melted using a transfer type plasma arc welding device with scanning speed of 1000 mm/min and welding current of 110 A. Laser re-melting and plasma arc re-melting samples with the gauge dimensions of 10 mm × 4 mm × 2 mm were cut using wire electro-discharge machining along the cross section.

The specimens were ground with 600~2500# sand papers and polished using a 0.025 μm silica suspension. The polished samples were etched with a 25% phosphoric acid solution in volume fraction to reveal the microstructure for observation. Microstructure analysis was performed on a ZEISS optical microscopy (OM) and FEI VERIOS 460 field emission scanning electron microscopy (SEM) equipped with an Oxford NORD LYS electron backscatter diffraction (EBSD) detector. The nanoindentation tests were conducted by a Hysitron TI 950 TriboIndenter equipped. The specimens were observed by EBSD following the typical EBSD preparation process and the scan step length was 0.1 μm.

## 3. Results and Discussion

As shown in Figure 1a, the three regions of the fusion zone, heat-affected zone, and substrate zone can be seen clearly in the laser re-melted sample. Good epitaxial growth was achieved in the fusion zone but small local stray grains existed in the region just below the top surface of the fusion zone, which is known as the columnar-to-equiaxed transition (CET), and stray grains can be removed with subsequent re-melting. The ratio of fusion zone depth to heat-affected zone depth was about 7. Grain boundaries and stray grains were eliminated for most parts of the melted region, and therefore, a single-crystal structure was formed. A fine dendrite structure with laser re-melting can be obviously seen, and there was no obvious solidification crack forming though the Al + Ti content in this alloy was relatively high [15]. The EBSD observation results (kernel average misorientation (KAM) and inverse pole figure (IPF)) in Figure 1b,c show the epitaxial growth of the single crystal to verify the same orientation of the melted zone and the substrate. As shown in the red-dashed box of Figure 1a, the EBSD observation zone includes three different zones: the fusion zone, heat-affected zone, and substrate zone. The KAM map shows the low average misorientation and only the crystallographic orientation of [001] exists in the IPF map. 

Figure 1d presents the optical images of the plasma arc re-melting of the three regions of the fusion zone, heat-affected zone, and the substrate zone, which can clearly be observed. Having an elliptical cross-section, the plasma arc re-melting zone is similar in shape to the laser re-melting zone, but a relatively low width/depth ratio results from different heat-source types. The laser energy distribution is a flat-top mode leading to a relatively great width/depth ratio, and the energy distribution of the plasma arc is an ellipsoidal heat source mode, resulting in a small width/depth ratio. The ratio of the fusion zone depth to the heat-affected zone depth was approximately 1.1. Stray grains were absent in the top of the fusion zone with the plasma arc re-melting. Consequently, it shows that the thermal input with the plasma arc re-melting process inside the fusion zone, especially the top and the bottom, was not much different and CET did not occur, but this also caused the heat-affected zone to be slightly large. The dendritic structure of the fusion zone was also fine and there were no obvious solidification cracks. Both the KAM and IPF figures in Figure 1e,f demonstrate the epitaxial growth of the single crystal with plasma arc repair.

The mechanical properties of Ni-based superalloys depend mainly on the morphology, size, distribution, and volume fraction of γ′ precipitates [19,20,21]. The laser re-melted and plasma arc re-melted DD407 samples were analyzed using SEM to characterize the microstructures. As indicated by the arrow in Figure 2g and Figure 3g, the spherical γ′ precipitates in the fusion zone were distributed homogeneously with average particle sizes in the range of 20 nm to approximately 40 nm in both repairing methods. In addition, larger γ′ precipitates in the fusion zone near the heat-affected zone, seen in Figure 2h, with radii of 30 nm to about 50 nm were observed. As shown in Figure 2i–k and in Figure 3i–k, most of the γ′ precipitate phases in the heat-affected zone were spherical with smaller size, and a few of the γ′ ones were cuboidal in shape with a larger size. The γ′ precipitates in the substrate region with a cuboidal shape had edge lengths of about 300 nm. 

As we can see in Figure 2 and Figure 3, the Vickers hardness reached the top in the fusion zone, which was related to small γ′ precipitates. However, the microhardness of laser re-melting zone was higher than that of the plasma arc re-melting zone, seen in Figure 2a–c and in Figure 3a–c, indicating the microstructure of the laser repairing was finer. In statistics, the microhardness of the fine γ′ precipitates zone was in the range of 440–460 HV. The microhardness of the coexisting region of the fine spherical γ’ phase and the larger cuboidal γ’ phase was 430–440 HV, as indicated in Figure 2d and Figure 3d,e. The microhardness of the large γ′ precipitates with only the cuboidal shape in the substrate zone was generally lower than 430 HV, as shown in Figure 2e,f and in Figure 3f.

Turbine blades are usually subject to blade tip abrasion. The fineness of the microstructure and the increase in hardness are advantageous for enhancing wear resistance [22]. But, the fine microstructure has a detrimental effect on high-temperature performance. Although this problem is not large in the blade tip area, local heat treatment is needed to regulate the γ′ precipitates to meet the actual need of blade repair.

The nanoindentation hardness (referred to as nanohardness) and Young’s modulus of the laser re-melted and plasma arc re-melted DD407 Ni-based superalloy samples were calculated with Oliver–Pharr’s method and the results are plotted in Figure 4b,d [23]. Figure 4a,c show the representative nanoindentation load–displacement curves of the single-crystal Ni-based superalloy samples. Note that a low displacement at peak load in Figure 4a,c indicates the high strength of the single-crystal Ni-based superalloy. It can be seen that the tendency of nanohardness was similar with that of Vickers hardness mentioned above, both in the laser re-melted and plasma arc re-melted DD407 superalloy samples, which clearly verifies that the results above are effective. The nanohardness of the laser re-melted DD407 superalloy, as seen in Figure 4a,b, reached the maximum value in the fusion zone and the nanohardness away from this zone gradually decreased. Similarly, the nanohardness of the DD407 superalloy sample with plasma arc re-melting hit the top in the heat-affected zone, and that of the other zones decreased significantly. There was no large difference between the nanohardness of the two re-melted samples in the fusion zone. Moreover, the maximum of the Young’s modulus for the laser re-melted DD407 superalloy sample can be seen in the fusion zone. The Young’s modulus of the DD407 superalloy with plasma arc re-melting had no obvious change tendency. In addition, both of the two repairing processes had a great Young’s modulus in the fusion zone.

## 4. Conclusions

The contrastive studies between laser repairing and plasma arc repairing were systematically investigated, and the main conclusions are as follows:Both of the laser repairing and plasma arc repairing processes can be used for single-crystal repair. In the two repairing processes, the fusion zone can maintain single-crystal epitaxial growth without significant cracks and shares a similar microstructure with fine γ′ precipitates.The heat-affected zone with laser re-melting was relatively small, while that with the plasma arc re-melting was relatively large. The CET phenomenon occurred at the top of the fusion zone with the laser re-melting, but it was not obvious at the top surface of the fusion zone with the plasma arc re-melting.The fusion zones in the two repairing processes had a similar nanohardness and Young’s modulus, but the fusion zone with laser re-melting had a relatively large microhardness.

The laser re-melting and plasma arc re-melting process can be used to eliminate defects such as tiny holes, microcracks, and freckles on the metal surface. The research in this paper mainly focused on the mechanisms of the repairing processes. Future research aims to establish an appropriate model and carry out powder feeding or powder-bed experiments on the basis of this paper, aiming at repairing the blade tip subjected to abrasion and cracks.

## Figures and Tables

**Figure 1 materials-12-01172-f001:**
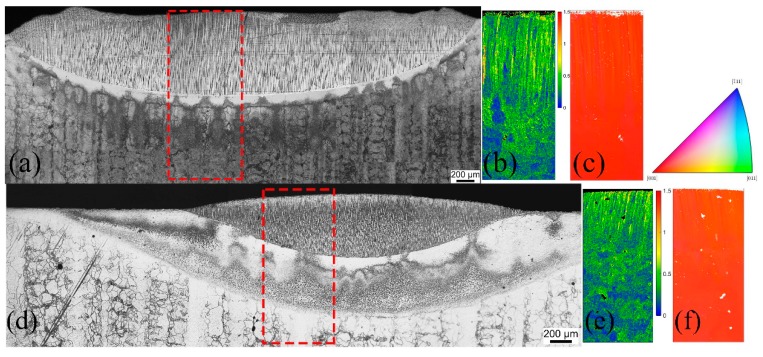
Optical micrographs of the transverse section with the (**a**) laser re-melted sample and (**d**) the plasma arc re-melted sample, which clearly show the fusion zone, heat-affected zone, and base metal. Corresponding images (**b**,**e**) are the kernel average misorientation (KAM) maps, and (**c**,**f**) are the inverse pole figure (IPF) maps, respectively. The red-dotted box in the optical images shows the electron backscatter diffraction (EBSD) observation region.

**Figure 2 materials-12-01172-f002:**
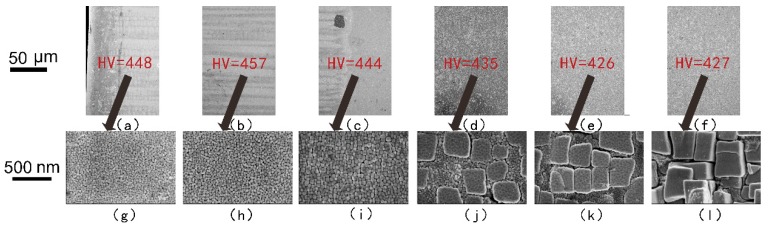
SEM micrographs showing the transverse section of the laser re-melted DD407 alloy in the (**a**,**b**) fusion zone, (**c**–**e**) heat-affected zone, and (**f**) substrate zone. (**g**–**l**) SEM images of high magnification in the center of the images above. The micro-HV data are highlighted in red.

**Figure 3 materials-12-01172-f003:**
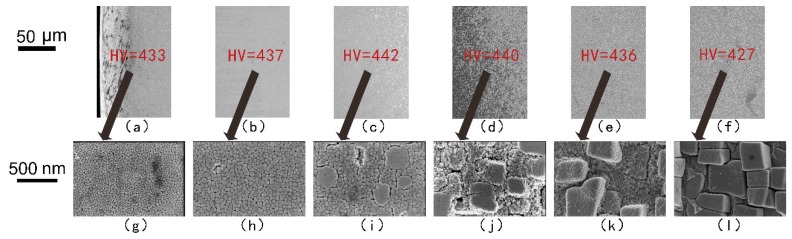
SEM micrographs showing the transverse section of the plasma arc re-melted DD407 alloy in the (**a**,**b**) fusion zone, (**c**–**e**) heat-affected zone, and (**f**) substrate zone. (**g**–**l**) SEM images of high magnification in the center of the images above. The micro-HV data are highlighted in red.

**Figure 4 materials-12-01172-f004:**
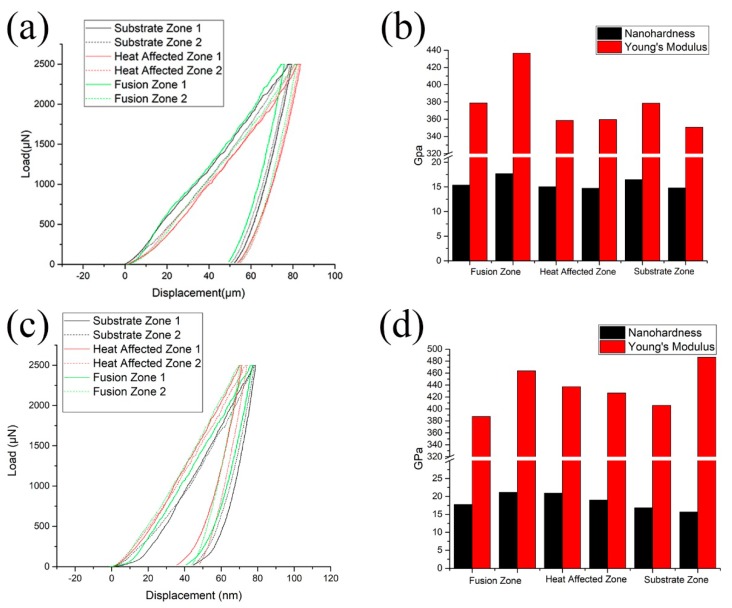
Nanoindentation curves of (**a**) the laser re-melted and (**c**) plasma arc re-melted DD407 superalloy samples within the fusion zone, heat-affected zone, and substrate zone, respectively. Their corresponding nanohardness and Young’s modulus of (**b**) the laser re-melted and (**d**) plasma arc re-melted DD407 superalloy samples.

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
