# Peer review of "Contrastive Studies between Laser Repairing and Plasma Arc Repairing on Single-Crystal Ni-Based Superalloy"

_materials, 2019, doi:10.3390/ma12071172_

Round 1

Reviewer 1 Report

Please, extent the chapter Conclusion with practical application of this method and direction of the next research

Author Response

Deat reviewer:

I am very grateful to your comments for the manuscript.We have studied comments carefully and have made correction which we hope meet with approval.

Reviewer 2 Report

The article entitled: „Contrastive Studies between Laser Repairing and  Plasma Arc Repairing on Single-crystal Ni-base  Superalloy” describes the differences in the microstrures of the Ni-base superalloy DD407 remelted using two methods, namely by laser technique and arc melting. The Authors found a substantial differences in the remelted regions between the two mentioned methods. I found this article interesting and worth publishing, however, during reading a following suggestions/comments arises as follows:

1.       Introduction: The article compare two methods: laser reparation and arc repairing. Since first method is relatively well described, then the second method is described in one short sentence, without mentioning any references. Please extend the Introduction by adding a wider description of arc repairing with references.

2.       Page 2, Line 66: The Authors presented nominal composition of the alloy and added a sentence about total Al+Ti amount in the alloy. The latest is unnecessary information since it is quite easy to calculate knowing Al and Ti concentration in the alloy. Please remove this part of the sentence.

3.       Page 3, line 96: Please explain “IPF” meaning, since it is used for the first time in the text.

4.       Page 3, Figure 1: The microstructure of the substrate shown in Fig. 1 (a) substantially differs from that shown in Fig 1 (d). Since the Authors declared that they studied the same material (DD407), then the question is why the microstructure of the substrates shown in Fig. 1 (a) and (d) are different?? Does this differences influence the remelting process? Please comment on this.

5.       Page 3, Figure 1: The images shown in Fig. 1 (a) and (d) are taken in a low magnification. An introduction of the images taken at higher magnification from each zone would surely help to illustrate the microstructures discussed in the text. Please add images at higher magnification.

6.       Page 4, lines 130 and 131: The Authors wrote: “The fineness of microstructure and the increase in hardness are advantageous for enhancing wear resistance”. Please add a proper reference or show the results supporting your assumption.

7.       Page 5, Conclusions: In the present form this is rather a summary of the observation rather than conclusions. Please re-write this chapter.

8.       General comment: The Authors mentioned, that both methods: laser and Arc can be used to repair the blare tips, which hurts due to the abrasion/erosion. In the article, remelting of the substrate was studied. Is it enough to remelt the broken blades tips? I think that the material lost due to erosion should be compensated, that means, that additional material should be “welded” on broken surface. This could result in introduction of porosity, pollutions etc. Why in the present study only remelting of bare material was studied then? Please comment on it and introduce a motivation for your study.

Moreover a several typing and/or language errors were found and marked (please find attached file).

Considering all above mentioned comments I would suggest to reconsider this article for publication after major revision.

For better understanding of my comments, please find attached a pdf file with hand-written comments.

Author Response

Deat reviewer:

I am very grateful to your comments for the manuscript. We have studied comments carefully and have made correction which we hope meet with approval. We have uploaded a Word file to response to the reviewer's comments.

Sincerely yours,

Xin Zhou

Reviewer 3 Report

This work on "Contrastive Studies…" could be considered for publication in <Materials> after a serious corrections and improvements are done. Many treatment methods may affect the results, so starting from the beginning, the aim of the paper should be clarified (moreover, if in the next Chapter you say that the "three zones… can be observed obviously"! _lines 98-99). The results of nanohardness measurement do not present a meaningful differences. 

No statistics is given in the paper, no final model presented. At the end, conclusions seem to be not fully supported. 

English language of this paper should be corrected (for reference, see the Enclosure with color markings). 

Author Response

(The authors gave the same response as above.)

Round 2

Reviewer 2 Report

The Authors seriously reacted on Reviewer comments, made necessary corrections and it can be published in Materials.